# PAC-Bayes-Empirical-Bernstein Inequality

**Ilya Tolstikhin**
Computing Centre
Russian Academy of Sciences
iliya.tolstikhin@gmail.com

**Yevgeny Seldin**
Queensland University of Technology
UC Berkeley
yevgeny.seldin@gmail.com

## Abstract

We present a PAC-Bayes-Empirical-Bernstein inequality. The inequality is based on a combination of the PAC-Bayesian bounding technique with an Empirical Bernstein bound. We show that when the empirical variance is significantly smaller than the empirical loss the PAC-Bayes-Empirical-Bernstein inequality is significantly tighter than the PAC-Bayes-kl inequality of Seeger (2002) and otherwise it is comparable. Our theoretical analysis is confirmed empirically on a synthetic example and several UCI datasets. The PAC-Bayes-Empirical-Bernstein inequality is an interesting example of an application of the PAC-Bayesian bounding technique to self-bounding functions.

## 1 Introduction

PAC-Bayesian analysis is a general and powerful tool for data-dependent analysis in machine learning. By now it has been applied in such diverse areas as supervised learning [1–4], unsupervised learning [4, 5], and reinforcement learning [6]. PAC-Bayesian analysis combines the best aspects of PAC learning and Bayesian learning: (1) it provides strict generalization guarantees (like VC-theory), (2) it is flexible and allows the incorporation of prior knowledge (like Bayesian learning), and (3) it provides data-dependent generalization guarantees (akin to Radamacher complexities).

PAC-Bayesian analysis provides concentration inequalities for the divergence between expected and empirical loss of randomized prediction rules. For a hypothesis space $\mathcal{H}$ a randomized prediction rule associated with a distribution $\rho$ over $\mathcal{H}$ operates by picking a hypothesis at random according to $\rho$ from $\mathcal{H}$ each time it has to make a prediction. If $\rho$ is a delta-distribution we recover classical prediction rules that pick a single hypothesis $h \in \mathcal{H}$. Otherwise, the prediction strategy resembles Bayesian prediction from the posterior distribution, with a distinction that $\rho$ does not have to be the Bayes posterior. Importantly, many of PAC-Bayesian inequalities hold for all posterior distributions $\rho$ simultaneously (with high probability over a random draw of a training set). Therefore, PAC-Bayesian bounds can be used in two ways. Ideally, we prefer to derive new algorithms that find the posterior distribution $\rho$ that minimizes the PAC-Bayesian bound on the expected loss. However, we can also use PAC-Bayesian bounds in order to estimate the expected loss of posterior distributions $\rho$ that were found by other algorithms, such as empirical risk minimization, regularized empirical risk minimization, Bayesian posteriors, and so forth. In such applications PAC-Bayesian bounds can be used to provide generalization guarantees for other methods and can be applied as a substitute for cross-validation in paratemer tuning (since the bounds hold for all posterior distributions $\rho$ simultaneously, we can apply the bounds to test multiple posterior distributions $\rho$ without suffering from over-fitting, in contrast with extensive applications of cross-validation).

There are two forms of PAC-Bayesian inequalities that are currently known to be the tightest depending on a situation. One is the PAC-Bayes-kl inequality of Seeger [7] and the other is the PAC-Bayes-Bernstein inequality of Seldin et. al. [8]. However, the PAC-Bayes-Bernstein inequality is expressed in terms of the true expected variance, which is rarely accessible in practice. Therefore, in order to apply the PAC-Bayes-Bernstein inequality we need an upper bound on the expected variance

(or, more precisely, on the average of the expected variances of losses of each hypothesis $h \in \mathcal{H}$ weighted according to the randomized prediction rule $\rho$). If the loss is bounded in the $[0, 1]$ interval the expected variance can be upper bounded by the expected loss and this bound can be used to recover the PAC-Bayes-kl inequality from the PAC-Bayes-Bernstein inequality (with slightly suboptimal constants and suboptimal behavior for small sample sizes). In fact, for the binary loss this result cannot be significantly improved (see Section 3). However, when the loss is not binary it may be possible to obtain a tighter bound on the variance, which will lead to a tighter bound on the loss than the PAC-Bayes-kl inequality. For example, in Seldin et. al. [6] a deterministic upper bound on the variance of importance-weighted sampling combined with PAC-Bayes-Bernstein inequality yielded an order of magnitude improvement relative to application of PAC-Bayes-kl inequality to the same problem. We note that the bound on the variance used by Seldin et. al. [6] depends on specific properties of importance-weighted sampling and does not apply to other problems.

In this work we derive the PAC-Bayes-Empirical-Bernstein bound, in which the expected average variance of the loss weighted by $\rho$ is replaced by the weighted average of the empirical variance of the loss. Bounding the expected variance by the empirical variance is generally tighter than bounding it by the empirical loss. Therefore, the PAC-Bayes-Empirical-Bernstein bound is generally tighter than the PAC-Bayes-kl bound, although the exact comparison also depends on the divergence between the posterior and the prior and the sample size. In Section 5 we provide an empirical comparison of the two bounds on several synthetic and UCI datasets.

The PAC-Bayes-Empirical-Bernstein bound is derived in two steps. In the first step we combine the PAC-Bayesian bounding technique with the Empirical Bernstein inequality [9] and derive a PAC-Bayesian bound on the variance. The PAC-Bayesian bound on the variance bounds the divergence between averages [weighted by $\rho$] of expected and empirical variances of the losses of hypotheses in $\mathcal{H}$ and holds with high probability for all averaging distributions $\rho$ simultaneously. In the second step the PAC-Bayesian bound on the variance is substituted into the PAC-Bayes-Bernstein inequality yielding the PAC-Bayes-Empirical-Bernstein bound.

The remainder of the paper is organized as follows. We start with some formal definitions and review the major PAC-Bayesian bounds in Section 2, provide our main results in Section 3 and their proof sketches in Section 4, and finish with experiments in Section 5 and conclusions in Section 6. Detailed proofs are provided in the supplementary material.

## 2 Problem Setting and Background

We start with providing the problem setting and then give some background on PAC-Bayesian analysis.

### 2.1 Notations and Definitions

We consider supervised learning setting with an input space $\mathcal{X}$, an output space $\mathcal{Y}$, an i.i.d. training sample $S = \{(X_i, Y_i)\}_{i=1}^{n}$ drawn according to an unknown distribution $\mathcal{D}$ on the product-space $\mathcal{X} \times \mathcal{Y}$, a loss function $\ell \colon \mathcal{Y}^2 \to [0, 1]$, and a hypothesis class $\mathcal{H}$. The elements of $\mathcal{H}$ are functions $h \colon \mathcal{X} \to \mathcal{Y}$ from the input space to the output space. We use $\ell_h(X, Y) = \ell(Y, h(X))$ to denote the loss of a hypothesis $h$ on a pair $(X, Y)$.

For a fixed hypothesis $h \in \mathcal{H}$ denote its expected loss by $L(h) = \mathbb{E}_{(X,Y)\sim\mathcal{D}}[\ell_h(X, Y)]$, the empirical loss $L_n(h) = \frac{1}{n}\sum_{i=1}^{n} \ell_h(X_i, Y_i)$, and the variance of the loss $\mathbb{V}(h) = \mathrm{Var}_{(X,Y)\sim\mathcal{D}}[\ell_h(X, Y)] = \mathbb{E}_{(X,Y)\sim\mathcal{D}}\left[\left(\ell_h(X, Y) - \mathbb{E}_{(X,Y)\sim\mathcal{D}}\left[\ell_h(X, Y)\right]\right)^2\right]$.

We define Gibbs regression rule $G_\rho$ associated with a distribution $\rho$ over $\mathcal{H}$ in the following way: for each point $X$ Gibbs regression rule draws a hypothesis $h$ according to $\rho$ and applies it to $X$. The expected loss of Gibbs regression rule is denoted by $L(G_\rho) = \mathbb{E}_{h\sim\rho}[L(h)]$ and the empirical loss is denoted by $L_n(G_\rho) = \mathbb{E}_{h\sim\rho}[L_n(h)]$. We use $\mathrm{KL}(\rho\|\pi) = \mathbb{E}_{h\sim\rho}\left[\ln\frac{\rho(h)}{\pi(h)}\right]$ to denote the Kullback-Leibler divergence between two probability distributions [10]. For two Bernoulli distributions with biases $p$ and $q$ we use $\mathrm{kl}(q\|p)$ as a shorthand for $\mathrm{KL}([q, 1 - q]\|[p, 1 - p])$. In the sequel we use $\mathbb{E}_\rho[\cdot]$ as a shorthand for $\mathbb{E}_{h\sim\rho}[\cdot]$.

## 2.2 PAC-Bayes-kl bound

Before presenting our results we review several existing PAC-Bayesian bounds. The result in Theorem 1 was presented by Maurer [11, Theorem 5] and is one of the tightest known concentration bounds on the expected loss of Gibbs regression rule. Theorem 1 generalizes (and slightly tightens) PAC-Bayes-kl inequality of Seeger [7, Theorem 1] from binary to arbitrary loss functions bounded in the $[0, 1]$ interval.

**Theorem 1.** *For any fixed probability distribution $\pi$ over $\mathcal{H}$, for any $n \geq 8$ and $\delta > 0$, with probability greater than $1 - \delta$ over a random draw of a sample $S$, for all distributions $\rho$ over $\mathcal{H}$ simultaneously:*

$$\mathrm{kl}\big(L_n(G_\rho)\|L(G_\rho)\big) \leq \frac{\mathrm{KL}(\rho\|\pi) + \ln\frac{2\sqrt{n}}{\delta}}{n}. \tag{1}$$

Since by Pinsker's inequality $|p - q| \leq \sqrt{\mathrm{kl}(q\|p)/2}$, Theorem 1 directly implies (up to minor factors) the more explicit PAC-Bayesian bound of McAllester [12]:

$$L(G_\rho) \leq L_n(G_\rho) + \sqrt{\frac{\mathrm{KL}(\rho\|\pi) + \ln\frac{2\sqrt{n}}{\delta}}{2n}}, \tag{2}$$

which holds with probability greater than $1 - \delta$ for all $\rho$ simultaneously. We note that kl is easy to invert numerically and for small values of $L_n(G_\rho)$ (less than $1/4$) the implicit bound in (1) is significantly tighter than the explicit bound in (2). This can be seen from another relaxation suggested by McAllester [2], which follows from (1) by the inequality $p \leq q + \sqrt{2q\mathrm{kl}(q\|p)} + 2\mathrm{kl}(q\|p)$ for $p < q$:

$$L(G_\rho) \leq L_n(G_\rho) + \sqrt{\frac{2L_n(G_\rho)\left(\mathrm{KL}(\rho\|\pi) + \ln\frac{2\sqrt{n}}{\delta}\right)}{n}} + \frac{2\left(\mathrm{KL}(\rho\|\pi) + \ln\frac{2\sqrt{n}}{\delta}\right)}{n}. \tag{3}$$

From inequality (3) we clearly see that inequality (1) achieves "fast convergence rate" or, in other words, when $L(G_\rho)$ is zero (or small compared to $1/\sqrt{n}$) the bound converges at the rate of $1/n$ rather than $1/\sqrt{n}$ as a function of $n$.

## 2.3 PAC-Bayes-Bernstein Bound

Seldin et. al. [8] introduced a general technique for combining PAC-Bayesian analysis with concentration of measure inequalities and derived the PAC-Bayes-Bernstein bound cited below. (The PAC-Bayes-Bernstein bound of Seldin et. al. holds for martingale sequences, but for simplicity in this paper we restrict ourselves to i.i.d. variables.)

**Theorem 2.** *For any fixed distribution $\pi$ over $\mathcal{H}$, for any $\delta_1 > 0$, and for any fixed $c_1 > 1$, with probability greater than $1 - \delta_1$ (over a draw of $S$) we have*

$$L(G_\rho) \leq L_n(G_\rho) + (1 + c_1)\sqrt{\frac{(e - 2)\mathbb{E}_\rho[\mathbb{V}(h)]\left(\mathrm{KL}(\rho\|\pi) + \ln\frac{\nu_1}{\delta_1}\right)}{n}} \tag{4}$$

*simultaneously for all distributions $\rho$ over $\mathcal{H}$ that satisfy*

$$\sqrt{\frac{\mathrm{KL}(\rho\|\pi) + \ln\frac{\nu_1}{\delta_1}}{(e - 2)\mathrm{E}_\rho[\mathbb{V}(h)]}} \leq \sqrt{n},$$

*where*

$$\nu_1 = \left\lceil \frac{1}{\ln c_1} \ln\left(\sqrt{\frac{(e - 2)n}{4\ln(1/\delta_1)}}\right)\right\rceil + 1,$$

*and for all other $\rho$ we have:*

$$L(G_\rho) \leq L_n(G_\rho) + 2\frac{\mathrm{KL}(\rho\|\pi) + \ln\frac{\nu_1}{\delta_1}}{n}.$$

*Furthermore, the result holds if $\mathbb{E}_\rho[\mathbb{V}(h)]$ is replaced by an upper bound $\bar{V}(\rho)$, as long as $\mathbb{E}_\rho[\mathbb{V}(h)] \leq \bar{V}(\rho) \leq \frac{1}{4}$ for all $\rho$.*

A few comments on Theorem 2 are in place here. First, we note that Seldin et. al. worked with cumulative losses and variances, whereas we work with normalized losses and variances, which means that their losses and variances differ by a multiplicative factor of $n$ from our definitions. Second, we note that the statement on the possibility of replacing $\mathbb{E}_\rho\left[\mathbb{V}(h)\right]$ by an upper bound is not part of [8, Theorem 8], but it is mentioned and analyzed explicitly in the text. The requirement that $\bar{V}(\rho) \leq \frac{1}{4}$ is not mentioned explicitly, but it follows directly from the necessity to preserve the relevant range of the trade-off parameter $\lambda$ in the proof of the theorem. Since $\frac{1}{4}$ is a trivial upper bound on the variance of a random variable bounded in the $[0, 1]$ interval, the requirement is not a limitation. Finally, we note that since we are working with "one-sided" variables (namely, the loss is bounded in the $[0, 1]$ interval rather than "two-sided" $[-1, 1]$ interval, which was considered in [8]) the variance is bounded by $\frac{1}{4}$ (rather than 1), which leads to a slight improvement in the value of $\nu_1$.

Since in reality we rarely have access to the expected variance $\mathbb{E}_\rho\left[\mathbb{V}(h)\right]$ the tightness of Theorem 2 entirely depends on the tightness of the upper bound $\bar{V}(\rho)$. If we use the trivial upper bound $\mathbb{E}_\rho\left[\mathbb{V}(h)\right] \leq \frac{1}{4}$ the result is roughly equivalent to (2), which is inferior to Theorem 1. Design of a tighter upper bound on $\mathbb{E}_\rho\left[\mathbb{V}(h)\right]$ that holds for all $\rho$ simultaneously is the subject of the following section.

## 3 Main Results

The key result of our paper is a PAC-Bayesian bound on the average expected variance $\mathbb{E}_\rho\left[\mathbb{V}(h)\right]$ given in terms of the average empirical variance $\mathbb{E}_\rho[\mathbb{V}_n(h)] = \mathbb{E}_{h\sim\rho}[\mathbb{V}_n(h)]$, where

$$\mathbb{V}_n(h) = \frac{1}{n-1}\sum_{i=1}^n\bigl(\ell_h(X_i, Y_i) - L_n(h)\bigr)^2 \tag{5}$$

is an unbiased estimate of the variance $\mathbb{V}(h)$. The bound is given in Theorem 3 and it holds with high probability for all distributions $\rho$ simultaneously. Substitution of this bound into Theorem 2 yields the PAC-Bayes-Empirical-Bernstein inequality given in Theorem 4. Thus, the PAC-Bayes-Empirical-Bernstein inequality is based on two subsequent applications of the PAC-Bayesian bounding technique.

### 3.1 PAC-Bayesian bound on the variance

Theorem 3 is based on an application of the PAC-Bayesian bounding technique to the difference $\mathbb{E}_\rho\left[\mathbb{V}(h)\right] - \mathbb{E}_\rho\left[\mathbb{V}_n(h)\right]$. We note that $\mathbb{V}_n(h)$ is a second-order U-statistics [13] and Theorem 3 provides an interesting example of combining PAC-Bayesian analysis with concentration inequalities for self-bounding functions.

**Theorem 3.** *For any fixed distribution $\pi$ over $\mathcal{H}$, any $c_2 > 1$ and $\delta_2 > 0$, with probability greater than $1 - \delta_2$ over a draw of $S$, for all distributions $\rho$ over $\mathcal{H}$ simultaneously:*

$$\mathbb{E}_\rho[\mathbb{V}(h)] \leq \mathbb{E}_\rho[\mathbb{V}_n(h)] + (1 + c_2)\sqrt{\frac{\mathbb{E}_\rho\left[\mathbb{V}_n(h)\right]\left(\mathrm{KL}(\rho\|\pi) + \ln\frac{\nu_2}{\delta_2}\right)}{2(n-1)}} + \frac{2c_2\left(\mathrm{KL}(\rho\|\pi) + \ln\frac{\nu_2}{\delta_2}\right)}{n-1}, \tag{6}$$

*where*

$$\nu_2 = \left\lceil \frac{1}{\ln c_2}\ln\left(\frac{1}{2}\sqrt{\frac{n-1}{\ln(1/\delta_2)} + 1} + \frac{1}{2}\right)\right\rceil.$$

Note that (6) closely resembles the explicit bound on $L(G_\rho)$ in (3). If the empirical variance $\mathbb{V}_n(h)$ is close to zero the impact of the second term of the bound (that scales with $1/\sqrt{n}$) is relatively small and we obtain "fast convergence rate" of $\mathbb{E}_\rho\left[\mathbb{V}_n(h)\right]$ to $\mathbb{E}_\rho\left[\mathbb{V}(h)\right]$. Finally, we note that the impact of $c_2$ on $\ln\nu_2$ is relatively small and so $c_2$ can be taken very close to 1.

### 3.2 PAC-Bayes-Empirical-Bernstein bound

Theorem 3 controls the average variance $\mathbb{E}_\rho[\mathbb{V}(h)]$ for all posterior distributions $\rho$ simultaneously. By taking $\delta_1 = \delta_2 = \frac{\delta}{2}$ we have the claims of Theorems 2 and 3 holding simultaneously with

probability greater than $1 - \delta$. Substitution of the bound on $\mathbb{E}_\rho[\mathbb{V}(h)]$ from Theorem 3 into the PAC-Bayes-Bernstein inequality in Theorem 2 yields the main result of our paper, the PAC-Bayes-Empirical-Bernstein inequality, that controls the loss of Gibbs regression rule $\mathbb{E}_\rho[L(h)]$ for all posterior distributions $\rho$ simultaneously.

**Theorem 4.** *Let $V_n(\rho)$ denote the right hand side of* (6) *(with $\delta_2 = \frac{\delta}{2}$) and let $\bar{V}_n(\rho) = \min\left(V_n(\rho), \frac{1}{4}\right)$. For any fixed distribution $\pi$ over $\mathcal{H}$, for any $\delta > 0$, and for any $c_1, c_2 > 1$, with probability greater than $1 - \delta$ (over a draw of S) we have:*

$$L(G_\rho) \le L_n(G_\rho) + (1 + c_1)\sqrt{\frac{(e-2)\bar{V}_n(\rho)\left(\mathrm{KL}(\rho\|\pi) + \ln\frac{2\nu_1}{\delta}\right)}{n}} \tag{7}$$

*simultaneously for all distributions $\rho$ over $\mathcal{H}$ that satisfy*

$$\sqrt{\frac{\mathrm{KL}(\rho\|\pi) + \ln\frac{2\nu_1}{\delta}}{(e-2)\bar{V}_n(\rho)}} \le \sqrt{n},$$

*where $\nu_1$ was defined in Theorem 2 (with $\delta_1 = \frac{\delta}{2}$), and for all other $\rho$ we have:*

$$L(G_\rho) \le L_n(G_\rho) + 2\frac{\mathrm{KL}(\rho\|\pi) + \ln\frac{2\nu_1}{\delta}}{n}.$$

Note that all the quantities in Theorem 4 are computable based on the sample.

As we can see immediately by comparing the $O(1/\sqrt{n})$ term in PAC-Bayes-Empirical-Bernstein inequality (PB-EB for brevity) with the corresponding term in the relaxed version of the PAC-Bayes-kl inequality (PB-kl for brevity) in equation (3), the PB-EB inequality can potentially be tighter when $\mathbb{E}_\rho[\mathbb{V}_n(h)] \le (1/(2(e-2)))L_n(G_\rho) \approx 0.7L_n(G_\rho)$. We also note that when the loss is bounded in the [0,1] interval we have $\mathbb{V}_n(h) \le (n/(n-1))L_n(h)$ (since $\ell_h(X,Y)^2 \le \ell_h(X,Y)$). Therefore, the PB-EB bound is never much worse than the PB-kl bound and if the empirical variance is small compared to the empirical loss it can be much tighter. We note that for the binary loss ($\ell(y,y') \in \{0,1\}$) we have $\mathbb{V}(h) = L(h)(1 - L(h))$ and in this case the empirical variance cannot be significantly smaller than the empirical loss and PB-EB does not provide an advantage over PB-kl. We also note that the unrelaxed version of the PB-kl inequality in equation (1) has better behavior for very small sample sizes and in such cases PB-kl can be tighter than PB-EB even when the empirical variance is small. To summarize the discussion, when $\mathbb{E}_\rho[\mathbb{V}_n(h)] \le 0.7L_n(G_\rho)$ the PB-EB inequality can be significantly tighter than the PB-kl bound and otherwise it is comparable (except for very small sample sizes). In Section 5 we provide a more detailed numerical comparison of the two inequalities.

## 4 Proofs

In this section we present a sketch of a proof of Theorem 3 and a proof of Theorem 4. Full details of the proof of Theorem 3 are provided in the supplementary material. The proof of Theorem 3 is based on the following lemma, which is at the base of all PAC-Bayesian theorems. (Since we could not find a reference, where the lemma is stated explicitly its proof is provided in the supplementary material.)

**Lemma 1.** *For any function $f_n : \mathcal{H} \times (\mathcal{X} \times \mathcal{Y})^n \to \mathbb{R}$ and for any distribution $\pi$ over $\mathcal{H}$, such that $\pi$ is independent of S, with probability greater than $1 - \delta$ over a random draw of S, for all distributions $\rho$ over $\mathcal{H}$ simultaneously:*

$$\mathbb{E}_\rho[f_n(h,S)] \le \mathrm{KL}(\rho\|\pi) + \ln\frac{1}{\delta} + \ln\mathbb{E}_\pi\left[\mathbb{E}_{S'\sim\mathcal{D}^n}\left[e^{f_n(h,S')}\right]\right]. \tag{8}$$

The smart part is to choose $f_n(h,S)$ so that we get the quantities of interest on the left hand side of (8) and at the same time are able to bound the last term on the right hand side of (8). Bounding of the moment generating function (the last term in (8)) is usually done by involving some known concentration of measure results. In the proof of Theorem 3 we use the fact that $n\mathbb{V}_n(h)$ satisfies the *self-bounding property* [14]. Specifically, for any $\lambda > 0$:

$$\mathbb{E}_{S\sim\mathcal{D}^n}\left[e^{\lambda(n\mathbb{V}(h)-n\mathbb{V}_n(h))-\frac{\lambda^2}{2}\frac{n^2}{n-1}\mathbb{V}(h)}\right] \le 1 \tag{9}$$

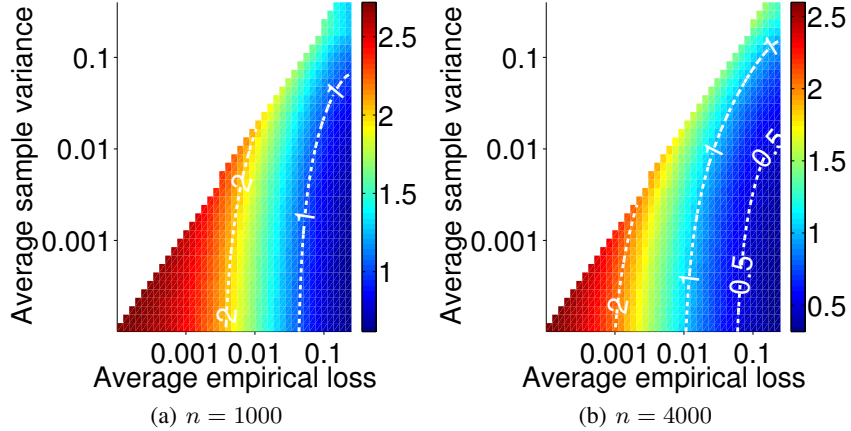

Figure 1: The Ratio of the gap between PB-EB and $L_n(G_\rho)$ to the gap between PB-kl and $L_n(G_\rho)$ for different values of $n$, $\mathbb{E}_\rho[\mathbb{V}_n(h)]$, and $L_n(G_\rho)$. PB-EB is tighter below the dashed line with label 1. The axes of the graphs are in log scale.

(see, for example, [9, Theorem 10]). We take $f_n(h,S) = \lambda\big(n\mathbb{V}(h) - n\mathbb{V}_n(h)\big) - \frac{\lambda^2}{2}\frac{n^2}{n-1}\mathbb{V}(h)$ and substitute $f_n$ and the bound on its moment generating function in (9) into (8). To complete the proof it is left to optimize the bound with respect to $\lambda$. Since it is impossible to minimize the bound simultaneously for all $\rho$ with a single value of $\lambda$, we follow the technique suggested by Seldin et. al. and take a grid of $\lambda$-s in a form of a geometric progression and apply a union bound over this grid. Then, for each $\rho$ we pick a value of $\lambda$ from the grid, which is the closest to the value of $\lambda$ that minimizes the bound for the corresponding $\rho$. (The approximation of the optimal $\lambda$ by the closest $\lambda$ from the grid is behind the factor $c_2$ in the bound and the $\ln \nu_2$ factor is the result of the union bound over the grid of $\lambda$-s.) Technical details of the derivation are provided in the supplementary material.

*Proof of Theorem 4.* By our choice of $\delta_1 = \delta_2 = \frac{\delta}{2}$ the upper bounds of Theorems 2 and 3 hold simultaneously with probability greater than $1 - \delta$. Therefore, with probability greater than $1 - \delta_2$ we have $\mathbb{E}_\rho\left[\mathbb{V}(h)\right] \leq \bar{V}_n(h) \leq \frac{1}{4}$ and the result follows by Theorem 2. $\square$

## 5 Experiments

Before presenting the experiments we present a general comparison of the behavior of the PB-EB and PB-kl bounds as a function of $L_n(G_\rho)$, $\mathbb{E}_\rho\left[\mathbb{V}_n(h)\right]$, and $n$. In Figure 1.a and 1.b we examine the ratio of the complexity parts of the two bounds

$$\frac{\text{PB-EB} - L_n(G_\rho)}{\text{PB-kl} - L_n(G_\rho)},$$

where PB-EB is used to denote the value of the PB-EB bound in equation (7) and PB-kl is used to denote the value of the PB-kl bound in equation (1). The ratio is presented in the $L_n(G_\rho)$ by $\mathbb{E}_\rho\left[\mathbb{V}_n(h)\right]$ plane for two values of $n$. In the illustrative comparison we took $\text{KL}(\rho\|\pi) = 18$ and in all the experiments presented in this section we take $c_1 = c_2 = 1.15$ and $\delta = 0.05$. As we wrote in the discussion of Theorem 4, PB-EB is never much worse than PB-kl and when $\mathbb{E}_\rho\left[\mathbb{V}_n(h)\right] \ll L_n(G_\rho)$ it can be significantly tighter. In the illustrative comparison in Figure 1, in the worst case the ratio is slightly above 2.5 and in the best case it is slightly above 0.3. We note that as the sample size grows the worst case ratio decreases (asymptotically down to 1.2) and the improvement of the best case ratio is unlimited.

As we already said, the advantage of the PB-EB inequality over the PB-kl inequality is most prominent in regression (for classification with zero-one loss it is roughly comparable to PB-kl). Below we provide regression experiments with $L_1$ loss on synthetic data and three datasets from the UCI repository [15]. We use the PB-EB and PB-kl bounds to bound the loss of a regularized empirical

risk minimization algorithm. In all our experiments the inputs $X_i$ lie in a $d$-dimensional unit ball centered at the origin ($\|X_i\|_2 \leq 1$) and the outputs $Y$ take values in $[-0.5, 0.5]$. The hypothesis class $\mathcal{H}_W$ is defined as

$$\mathcal{H}_W = \left\{ h_{\boldsymbol{w}}(X) = \langle \boldsymbol{w}, X \rangle : \boldsymbol{w} \in \mathbb{R}^d, \|\boldsymbol{w}\|_2 \leq 0.5 \right\}.$$

This construction ensures that the $L_1$ regression loss $\ell(y, y') = |y - y'|$ is bounded in the $[0, 1]$ interval. We use uniform prior distribution over $\mathcal{H}_W$ defined by $\pi(\boldsymbol{w}) = \left( V(1/2, d) \right)^{-1}$, where $V(r, d)$ is the volume of a $d$-dimensional ball with radius $r$. The posterior distribution $\rho_{\hat{\boldsymbol{w}}}$ is taken to be a uniform distribution on a $d$-dimensional ball of radius $\epsilon$ centered at the weight vector $\hat{\boldsymbol{w}}$, where $\hat{\boldsymbol{w}}$ is the solution of the following minimization problem:

$$\hat{\boldsymbol{w}} = \arg\min_{\boldsymbol{w}} \frac{1}{n} \sum_{i=1}^{n} |Y_i - \langle \boldsymbol{w}, X_i \rangle| + \lambda^* \|\boldsymbol{w}\|_2^2. \tag{10}$$

Note that (10) is a quadratic program and can be solved by various numerical solvers (we used Matlab quadprog). The role of the regularization parameter $\lambda^* \|\boldsymbol{w}\|_2^2$ is to ensure that the posterior distribution is supported by $\mathcal{H}_W$. We use binary search in order to find the minimal (non-negative) $\lambda^*$, such that the posterior $\rho_{\hat{\boldsymbol{w}}}$ is supported by $\mathcal{H}_W$ (meaning that the ball of radius $\epsilon$ around $\hat{\boldsymbol{w}}$ is within the ball of radius $0.5$ around the origin). In all the experiments below we used $\epsilon = 0.05$.

## 5.1 Synthetic data

Our synthetic datasets are produced as follows. We take inputs $X_1, \ldots, X_n$ uniformly distributed in a $d$-dimensional unit ball centered at the origin. Then we define

$$Y_i = \sigma_0 \left( 50 \cdot \langle \boldsymbol{w}_0, X_i \rangle \right) + \epsilon_i$$

with weight vector $\boldsymbol{w}_0 \in \mathbb{R}^d$, centred sigmoid function $\sigma_0(z) = \frac{1}{1+e^{-z}} - 0.5$ which takes values in $[-0.5, 0.5]$, and noise $\epsilon_i$ independent of $X_i$ and uniformly distributed in $[-a_i, a_i]$ with

$$a_i = \begin{cases} \min\left(0.1, 0.5 - \sigma_0(50 \cdot \langle \boldsymbol{w}_0, X_i \rangle)\right), & \text{for } \sigma_0(50 \cdot \langle \boldsymbol{w}_0, X_i \rangle) \geq 0; \\ \min\left(0.1, 0.5 + \sigma_0(50 \cdot \langle \boldsymbol{w}_0, X_i \rangle)\right), & \text{for } \sigma_0(50 \cdot \langle \boldsymbol{w}_0, X_i \rangle) < 0. \end{cases}$$

This design ensures that $Y_i \in [-0.5, 0.5]$. The sigmoid function creates a mismatch between the data generating distribution and the linear hypothesis class. Together with relatively small level of the noise ($\epsilon_i \leq 0.1$) this results in small empirical variance of the loss $\mathbb{V}_n(h)$ and medium to high empirical loss $L_n(h)$. Let us denote the $j$-th coordinate of a vector $\boldsymbol{u} \in \mathbb{R}^d$ by $\boldsymbol{u}^j$ and the number of nonzero coordinates of $\boldsymbol{u}$ by $\|\boldsymbol{u}\|_0$. We choose the weight vector $\boldsymbol{w}_0$ to have only a few nonzero coordinates and consider two settings. In the first setting $d \in \{2, 5\}$, $\|\boldsymbol{w}_0\|_0 = 2$, $\boldsymbol{w}_0^1 = 0.12$, and $\boldsymbol{w}_0^2 = -0.04$ and in the second setting $d \in \{3, 6\}$, $\|\boldsymbol{w}_0\|_0 = 3$, $\boldsymbol{w}_0^1 = -0.08$, $\boldsymbol{w}_0^2 = 0.05$, and $\boldsymbol{w}_0^3 = 0.2$.

For each sample size ranging from 300 to 4000 we averaged the bounds over 10 randomly generated datasets. The results are presented in Figure 2. We see that except for very small sample sizes ($n < 1000$) the PB-EB bound outperforms the PB-kl bound. Inferior performance for very small sample sizes is a result of domination of the $O(1/n)$ term in the PB-EB bound (7). As soon as $n$ gets large enough this term significantly decreases and PB-EB dominates PB-kl.

## 5.2 UCI datasets

We compare our PAC-Bayes-Empirical-Bernstein inequality (7) with the PAC-Bayes-kl inequality (1) on three UCI regression datasets: Wine Quality, Parkinsons Telemonitoring, and Concrete Compressive Strength. For each dataset we centred and normalised both outputs and inputs so that $Y_i \in [-0.5, 0.5]$ and $\|X_i\| \leq 1$. The results for 5-fold train-test split of the data together with basic descriptions of the datasets are presented in Table 1.

## 6 Conclusions and future work

We derived a new PAC-Bayesian bound that controls the convergence of averages of empirical variances of losses of hypotheses in $\mathcal{H}$ to averages of expected variances of losses of hypothesis in $\mathcal{H}$ simultaneously for all averaging distributions $\rho$. This bound is an interesting example of combination

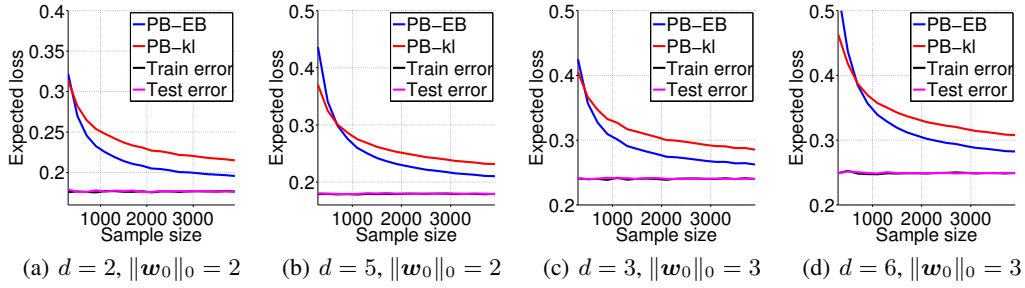

(a) $d = 2, \|\boldsymbol{w}_0\|_0 = 2$  (b) $d = 5, \|\boldsymbol{w}_0\|_0 = 2$  (c) $d = 3, \|\boldsymbol{w}_0\|_0 = 3$  (d) $d = 6, \|\boldsymbol{w}_0\|_0 = 3$

Figure 2: The values of the PAC-Bayes-kl and PAC-Bayes-Empirical-Bernstein bounds together with the test and train errors on synthetic data. The values are averaged over the 10 random draws of training and test sets.

Table 1: Results for the UCI datasets

| Dataset | $n$ | $d$ | Train | Test | PB-kl bound | PB-EB bound |
|---|---|---|---|---|---|---|
| winequality | 6497 | 11 | $0.106 \pm 0.0005$ | $0.106 \pm 0.0022$ | $0.175 \pm 0.0006$ | $\mathbf{0.162 \pm 0.0006}$ |
| parkinsons | 5875 | 16 | $0.188 \pm 0.0014$ | $0.188 \pm 0.0055$ | $0.266 \pm 0.0013$ | $\mathbf{0.250 \pm 0.0012}$ |
| concrete | 1030 | 8 | $0.110 \pm 0.0008$ | $0.111 \pm 0.0038$ | $\mathbf{0.242 \pm 0.0010}$ | $0.264 \pm 0.0011$ |

of PAC-Bayesian bounding technique with concentration inequalities for self-bounding functions. We applied the bound to derive the PAC-Bayes-Empirical-Bernstein inequality which is a powerful Bernstein-type inequality outperforming the state-of-the-art PAC-Bayes-kl inequality of Seeger [7] in situations, where the empirical variance is smaller than the empirical loss and otherwise comparable to PAC-Bayes-kl. We also demonstrated an empirical advantage of the new PAC-Bayes-Empirical-Bernstein inequality over the PAC-Bayes-kl inequality on several synthetic and real-life regression datasets.

Our work opens a number of interesting directions for future research. One of the most important of them is to derive algorithms that will directly minimize the PAC-Bayes-Empirical-Bernstein bound. Another interesting direction would be to decrease the last term in the bound in Theorem 3, as it is done in the PAC-Bayes-kl inequality. This can probably be achieved by deriving a PAC-Bayes-kl inequality for the variance.

### Acknowledgments

The authors are thankful to Anton Osokin for useful discussions and to the anonymous reviewers for their comments. This research was supported by an Australian Research Council Australian Laureate Fellowship (FL110100281) and a Russian Foundation for Basic Research grants 13-07-00677, 14-07-00847.

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
