[Supplementary Material]

# PAC-Bayes-Empirical-Bernstein Inequality

*Full Version Including Appendices*

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

## A    Proof of Theorem 3

First, we are going to use the so-called *self-bounding* property [14] of the random variable $\mathbb{V}_n(h)$ to derive a tight bound on the moment generating function of the difference $\mathbb{V}(h) - \mathbb{V}_n(h)$. It is done by using the following result which is an intermediate step in the proof of a concentration inequality for self-bounding functions presented in [16, Theorem 13]. The result is given in the forth line before the end of the proof of [16, Theorem 13].

**Theorem 5.** *Let $\boldsymbol{X} = (X_1, \ldots, X_n)$ be a vector of independent random variables with values in some set $\mathcal{X}$. For $1 \le k \le n$ and $x \in \mathcal{X}$, we will write $\boldsymbol{X}_{k,x}$ to denote the vector obtained from $\boldsymbol{X}$ by replacing $X_k$ by $x$. Suppose that $a \ge 1$ and that $Z = Z(\boldsymbol{X})$ is a random variable $Z : \mathcal{X}^n \to \mathbb{R}$ that satisfies*

$$\forall k: \ Z(\boldsymbol{X}) - \inf_{x \in \mathcal{X}} Z(\boldsymbol{X}_{k,x}) \le 1, \tag{11}$$

$$\sum_{k=1}^n \big( Z(\boldsymbol{X}) - \inf_{x \in \mathcal{X}} Z(\boldsymbol{X}_{k,x}) \big)^2 \le a Z(\boldsymbol{X}) \tag{12}$$

*almost surely. Then for $s > 0$*

$$\mathbb{E}\left[ e^{s(\mathbb{E}[Z] - Z)} \right] \le e^{a s^2 \mathbb{E}[Z]/2}. \tag{13}$$

We will also need the following simple result:

**Lemma 2.** *For any finite sequence of real numbers $\{x_1, \ldots, x_n\}$ the following holds:*

$$\frac{1}{n(n-1)} \sum_{1 \le i < j \le n} (x_i - x_j)^2 = \frac{1}{n-1} \sum_{i=1}^{n} \left( x_i - \frac{1}{n} \sum_{j=1}^{n} x_j \right)^2.$$

*Proof.*

$$\frac{1}{n-1} \sum_{i=1}^{n} \left( x_i - \frac{1}{n} \sum_{j=1}^{n} x_j \right)^2 = \frac{1}{n-1} \sum_{i=1}^{n} \left( x_i^2 - \frac{2}{n} x_i \sum_{j=1}^{n} x_j + \frac{1}{n^2} \left( \sum_{j=1}^{n} x_j \right)^2 \right)$$

$$= \frac{1}{n-1} \left( \sum_{i=1}^{n} x_i^2 - \frac{2}{n} \sum_{i=1}^{n} x_i \sum_{j=1}^{n} x_j + \frac{1}{n^2} \sum_{i=1}^{n} \left( \sum_{j=1}^{n} x_j \right)^2 \right)$$

$$= \frac{1}{n-1} \left( \sum_{i=1}^{n} x_i^2 - \frac{1}{n} \left( \sum_{j=1}^{n} x_j \right)^2 \right)$$

$$= \frac{1}{n(n-1)} \left( (n-1) \sum_{i=1}^{n} x_i^2 - 2 \sum_{1 \le i < j \le n} x_i x_j \right)$$

$$= \frac{1}{n(n-1)} \sum_{1 \le i < j \le n} (x_i - x_j)^2.$$

$\square$

*Proof of Theorem 3.* It is proved in [9, Theorem 10] that the random variable $n\mathbb{V}_n(h)$ satisfies conditions (11) and (12) with $a = \frac{n}{n-1}$. Hence, by (13) for any $\lambda > 0$ we obtain

$$\mathbb{E}\left[ e^{\lambda(n\mathbb{V}(h) - n\mathbb{V}_n(h))} \right] \le e^{\frac{\lambda^2}{2} \frac{n^2}{n-1} \mathbb{V}(h)}$$

or, equivalently,

$$\mathbb{E}\left[ e^{\lambda(n\mathbb{V}(h) - n\mathbb{V}_n(h)) - \frac{\lambda^2}{2} \frac{n^2}{n-1} \mathbb{V}(h)} \right] \le 1, \tag{14}$$

which is a bound on the moment generating function of the random variable

$$\Phi_\lambda(h) = \lambda n \left( 1 - \frac{\lambda n}{2(n-1)} \right) \mathbb{V}(h) - \lambda n \mathbb{V}_n(h).$$

By substituting $\Phi_\lambda(h)$ into Lemma 1 we obtain that for $\pi$ that is independent of the data, with probability greater than $1 - \delta$ for all distributions $\rho$ simultaneously:

$$\mathbb{E}_\rho[\Phi_\lambda(h)] \le \mathrm{KL}(\rho\|\pi) + \ln \frac{1}{\delta},$$

or

$$\left( 1 - \frac{\lambda n}{2(n-1)} \right) \mathbb{E}_\rho[\mathbb{V}(h)] \le \mathbb{E}_\rho[\mathbb{V}_n(h)] + \frac{\mathrm{KL}(\rho\|\pi) + \ln \frac{1}{\delta}}{\lambda n}.$$

By assuming that $\lambda \le \frac{2(n-1)}{n}$ and dividing both sides of the inequality by $1 - \frac{\lambda n}{2(n-1)}$ we obtain:

$$\mathbb{E}_\rho[\mathbb{V}(h)] \le \frac{\mathbb{E}_\rho[\mathbb{V}_n(h)]}{\left( 1 - \frac{\lambda n}{2(n-1)} \right)} + \frac{\mathrm{KL}(\rho\|\pi) + \ln \frac{1}{\delta}}{\lambda n \left( 1 - \frac{\lambda n}{2(n-1)} \right)}. \tag{15}$$

Note that the right hand side of (15) cannot be minimized simultaneously for all $\rho$ by a single value of $\lambda$. In the remainder of the proof we first find the optimal value of $\lambda$ that minimizes (15) and then design a grid of $\lambda$-s in a form of a geometric progression and approximate the optimal $\lambda$ by the nearest $\lambda$ from the grid. This bounding technique is inspired by [8].

Let us introduce the following notations:

$$t = \frac{\lambda n}{2(n-1)}, \quad a = \mathbb{E}_\rho[\mathbb{V}_n(h)], \quad b = \frac{\mathrm{KL}(\rho\|\pi) + \ln\frac{1}{\delta}}{2(n-1)}. \tag{16}$$

Then we can rewrite (15) as:

$$\mathbb{E}_\rho[\mathbb{V}(h)] \leq F(t) = \frac{a}{1-t} + \frac{b}{t(1-t)}, \tag{17}$$

where $a, b \geq 0$ and $0 < t \leq 1$ since we assumed that $\lambda \leq \frac{2(n-1)}{n}$.

Note that for $t \in (0, 1]$

$$\frac{\partial F}{\partial t} = \frac{a}{(1-t)^2} - \frac{(1-2t)b}{t^2(1-t)^2},$$

$$\frac{\partial^2 F}{\partial t^2} = \frac{2a}{(1-t)^3} + \frac{2bt^2(1-t)^2 + 2b(2t-1)^2(1-t)t}{t^4(1-t)^4} \geq 0,$$

and, therefore, $F(t)$ is convex on the interval of interest and achieves its minimum at the positive solution of

$$at^2 + 2bt - b = 0,$$

which is

$$t^* = \frac{\sqrt{b^2 + ab} - b}{a} = \frac{\sqrt{b}(\sqrt{a+b} - \sqrt{b})}{(a+b) - b} = \frac{\sqrt{b}}{\sqrt{a+b} + \sqrt{b}} = \frac{1}{\sqrt{a/b+1} + 1} \leq \frac{1}{2}. \tag{18}$$

Now we are going to cover the relevant interval of $t$-s by a geometrically spaced sequence of $(t_k)_{k \in \mathbb{N}+}$. We have already obtained an upper bound on the relevant interval in (18). For the lower bound we substitute the values of $a$ and $b$ into (18) and obtain

$$t^* = \left( \sqrt{\frac{2(n-1)\mathbb{E}_\rho[\mathbb{V}_n(h)]}{\mathrm{KL}(\rho\|\pi) + \ln 1/\delta}} + 1 + 1 \right)^{-1}.$$

Considering the fact that $\mathrm{KL}(\rho\|\pi) \geq 0$ and $\mathbb{V}_n(h) \leq \frac{1}{2}$ (which is a simple consequence of Lemma 2 and our assumption that the loss is bounded in the $[0, 1]$ interval) we have

$$t^* \geq \left( \sqrt{\frac{n-1}{\ln 1/\delta}} + 1 + 1 \right)^{-1}.$$

Therefore, the range of $t$ we are interested in is

$$t \in \left[ \left( \sqrt{\frac{n-1}{\ln 1/\delta}} + 1 + 1 \right)^{-1}, \frac{1}{2} \right].$$

We cover the above range with the following sequence of $t$-s: $t_i = c^i \left( \sqrt{\frac{n-1}{\ln 1/\delta}} + 1 + 1 \right)^{-1}$, $i = 0, \ldots, m-1$, where $c > 1$. It suffices to take

$$m = \left\lceil \frac{1}{\ln c} \ln \left( \frac{1}{2} \sqrt{\frac{n-1}{\ln 1/\delta}} + 1 + \frac{1}{2} \right) \right\rceil$$

in order to cover the relevant interval. The value $t_{m-1}$ is the last value that is strictly less than $\frac{1}{2}$. For any particular training set we can find the value $t_{i^*}$, $i^* \in \{0, \ldots, m-1\}$, which satisfies

$$t_{i^*} \leq t^* \leq t_{i^*+1} \leq ct_{i^*},$$

where $t^*$ is the optimal value that minimizes the r.h.s. of (17) for a given $\rho$. Using this fact we get

$$
\begin{aligned}
F(t_{i^*}) &= \frac{a}{1 - t_{i^*}} + \frac{b}{t_{i^*}(1 - t_{i^*})} \\
&\leq \frac{a}{1 - t^*} + \frac{b}{(t^*/c)(1 - t^*)} \\
&= \frac{a}{1 - \frac{\sqrt{b}}{\sqrt{b} + \sqrt{a+b}}} + \frac{cb}{\left(1 - \frac{\sqrt{b}}{\sqrt{b} + \sqrt{a+b}}\right)\frac{\sqrt{b}}{\sqrt{b} + \sqrt{a+b}}} \\
&= \frac{a + cb + c\sqrt{b(a+b)}}{1 - \frac{\sqrt{b}}{\sqrt{b} + \sqrt{a+b}}} \\
&= \frac{\left(a + cb + c\sqrt{b(a+b)}\right)\left(\sqrt{a+b} + \sqrt{b}\right)}{\sqrt{a+b}} \\
&= a + cb + c\sqrt{b(a+b)} + \frac{a\sqrt{b}}{\sqrt{a+b}} + \frac{cb\sqrt{b}}{\sqrt{a+b}} + cb \\
&\leq a + (1+c)\sqrt{ab} + 4cb,
\end{aligned}
\tag{19}
$$

where in the last inequality we used the fact that $\sqrt{a+b} \leq \sqrt{a} + \sqrt{b}$. Substitution of the values of $a$ and $b$ yields (6) $\qquad\square$

## B  Appendix to Section 5

Here we provide proofs of results from section 5 and some technical discussion. First we will need the following result.

**Lemma 3.** *Consider the random variable $\xi = \langle \boldsymbol{w}, \boldsymbol{v} \rangle$ where $\boldsymbol{w} \in \mathbb{R}^d$ is distributed uniformly over the d-dimentional ball of radius $\epsilon$ centred at the origin and $\boldsymbol{v} \in \mathbb{R}^d$ is a fixed vector with nonzero and finite euclidian norm $0 < \|\boldsymbol{v}\|_2 \leq \infty$. Then random variable $\xi$ has the following density function $p_\xi(x)$ with finite support $[-\epsilon\|\boldsymbol{v}\|_2, \epsilon\|\boldsymbol{v}\|_2]$:*

$$
p_\xi(x) = \frac{\left(1 - \frac{x^2}{\epsilon^2\|\boldsymbol{v}\|_2^2}\right)^{\frac{d-1}{2}}}{N(\boldsymbol{v}, \epsilon, d)}
$$

*where*

$$
N(\boldsymbol{v}, \epsilon, d) = 2\,\epsilon\|\boldsymbol{v}\|_2 \int_0^{\frac{\pi}{2}} \cos^d(t)\,dt.
$$

*Also*

$$
\mathbb{E}[\xi] = 0, \quad \mathbb{V}[\xi] = \frac{(\epsilon\|\boldsymbol{v}\|_2)^2}{d+2}.
$$

*Proof.* First of all note that $\xi \in [-\epsilon\|\boldsymbol{v}\|_2, \epsilon\|\boldsymbol{v}\|_2]$ which is the consequence of Cauchy–Schwarz inequality. Also dew to the symmetry of the support of $\boldsymbol{w}$ we can restrict ourselves to the situation when $\boldsymbol{v}$ has only one nonzero coordinate which we'll assume to be the first cordinate.

Let us denote $j$-th coordinate of a vector $\boldsymbol{u} \in \mathbb{R}^d$, $j = 1, \ldots, d$, using the upper index $\boldsymbol{u}^j$. Then for any value $C$ from the support of $p_\xi$ condition $\xi = C$ is equivalent to $\boldsymbol{w}^1 = C/\boldsymbol{v}^1$ and restricts $\boldsymbol{w}$ to lie in the $(d-1)$-dimensional ball of radius $\sqrt{\epsilon^2 - (C/\|\boldsymbol{v}\|_2)^2}$ centred at the origin. Then it is obvious that

$$
p_\xi(x) \propto \left(1 - \frac{x^2}{\epsilon^2\|\boldsymbol{v}\|_2^2}\right)^{\frac{d-1}{2}}.
$$

Now it suffices to find the normalizing constant which we'll denote $N(\boldsymbol{v}, \epsilon, d)$:

$$
N(\boldsymbol{v}, \epsilon, d) = \int_{-\epsilon\|\boldsymbol{v}\|_2}^{\epsilon\|\boldsymbol{v}\|_2} \left(1 - \frac{x^2}{\epsilon^2\|\boldsymbol{v}\|_2^2}\right)^{\frac{d-1}{2}} dx = 2\int_0^{\epsilon\|\boldsymbol{v}\|_2} \left(1 - \frac{x^2}{\epsilon^2\|\boldsymbol{v}\|_2^2}\right)^{\frac{d-1}{2}} dx.
$$

Denoting $\sin(t) = \frac{x}{\epsilon \|v\|_2}$ we get

$$N(\boldsymbol{v}, \epsilon, d) = 2\,\epsilon\|\boldsymbol{v}\|_2 \int_0^{\frac{\pi}{2}} \cos^d(t)dt,$$

which completes the proof of the first part of lemma.

Equation $\mathbb{E}[\xi] = 0$ follows from the fact that $\xi$ has symmetric distribution. Finally we use the following reduction formula to compute the variance $\mathbb{V}[\xi]$. It states that for any $m, n \in \mathbb{N}$

$$\int \sin^m(t)\cos^n(t)dt = -\frac{\sin^{m-1}(t)\cos^{n+1}(t)}{m+n} + \frac{m-1}{m+n}\int \sin^{m-2}(t)\cos^n(t)dt. \qquad (20)$$

Since

$$\mathbb{V}[\xi] = \frac{2\int_0^{\epsilon\|\boldsymbol{v}\|_2} x^2\left(1 - \frac{x^2}{\epsilon^2\|\boldsymbol{v}\|_2^2}\right)^{\frac{d-1}{2}} dx}{N(\boldsymbol{v}, \epsilon, d)},$$

again denoting $\sin(t) = \frac{x}{\epsilon\|v\|_2}$ we get

$$\mathbb{V}[\xi] = \frac{2(\epsilon\|\boldsymbol{v}\|_2)^3}{N(\boldsymbol{v}, \epsilon, d)}\int_0^{\frac{\pi}{2}} \sin^2(t)\cos^d(t)dt.$$

Using reduction formula (20) we conclude that

$$\mathbb{V}[\xi] = \frac{2(\epsilon\|\boldsymbol{v}\|_2)^3}{N(\boldsymbol{v}, \epsilon, d)}\frac{1}{d+2}\int_0^{\frac{\pi}{2}} \cos^d(t)dt = \frac{(\epsilon\|\boldsymbol{v}\|_2)^2}{d+2}.$$

$\square$

Note that it is easy to recursively compute $N(\boldsymbol{v}, \epsilon, d)$ using the following reduction formula:

$$\int \cos^d(t)dt = \frac{1}{d}\cos^{d-1}(t)\sin(t) + \frac{d-1}{d}\int \cos^{d-2}(t)dt.$$

Now we are ready to derive all the quantities appearing in the PAC-Bayes bounds for our experimental setting. We will begin with the following result, which holds only for the particular choice of the radius $\epsilon$ of the posterior distribution.

**Theorem 6.** *Let the posterior and prior distributions $\rho_{\hat{\boldsymbol{w}}}$ and $\pi$ be defined as in Section 5, take the radius of posterior distribution to be $\hat{\epsilon} = \min_{i=1,\ldots,n} |Y_i - \langle \hat{\boldsymbol{w}}, X_i \rangle|$, and assume that $\hat{\epsilon} > 0$. Then we have*

$$\mathrm{KL}(\rho_{\hat{\boldsymbol{w}}}\|\pi) = d\ln\frac{2}{\hat{\epsilon}}; \qquad (21)$$

$$\mathbb{E}_{h\sim\rho_{\hat{\boldsymbol{w}}}}[L_n(h)] = L_n(h_{\hat{\boldsymbol{w}}}); \qquad (22)$$

$$\mathbb{E}_{h\sim\rho_{\hat{\boldsymbol{w}}}}[\mathbb{V}_n(h)] = \frac{1}{n-1}\sum_{i=1}^n\left((Y_i - \langle\hat{\boldsymbol{w}}, X_i\rangle)^2 + \frac{\hat{\epsilon}^2}{d+2}\|X_i\|_2^2\right) - \frac{n}{n-1}\left(L_n(h_{\hat{\boldsymbol{w}}})\right)^2 -$$

$$- \frac{\hat{\epsilon}^2}{4n(n-1)(d+2)}\sum_{i=1}^n\sum_{j=1}^n\langle X_i, X_j\rangle \mathrm{sgn}\{(Y_i - \langle\hat{\boldsymbol{w}}, X_i\rangle)(Y_j - \langle\hat{\boldsymbol{w}}, X_j\rangle)\}. \qquad (23)$$

*Proof.* Let us start from the derivation of (21):

$$\mathrm{KL}(\rho\|\pi) = \int_{\|\boldsymbol{w}\|\leq\frac{1}{2}}\rho_{\hat{\boldsymbol{w}}}(\boldsymbol{w})\ln\frac{\rho_{\hat{\boldsymbol{w}}}(\boldsymbol{w})}{\pi(w)}d\boldsymbol{w} =$$

$$= \int_{\|\boldsymbol{w}\|\leq\frac{1}{2}}\mathbb{1}\{\|\boldsymbol{w} - \hat{\boldsymbol{w}}\|_2 \leq \hat{\epsilon}\}\frac{1}{V(\hat{\epsilon}, d)}\ln\frac{V(1/2, d)}{V(\hat{\epsilon}, d)}d\boldsymbol{w} = d\ln\frac{2}{\hat{\epsilon}},$$

where $V(\epsilon, d)$ is the volume of $d$-dimensional ball with radius $\epsilon$.

Now recall the definition
$$\hat{\epsilon} = \min_{i=1,\dots,n} \left( |Y_i - \langle \hat{\boldsymbol{w}}, X_i \rangle| \right).$$
It implies that for any $i = 1, \dots, n$ random variables $\xi_i = Y_i - \langle \boldsymbol{w}, X_i \rangle$ (where $\boldsymbol{w} \sim \rho_{\hat{\boldsymbol{w}}}$) do not change their signs. Then equation (22) follows immediately from the definition:

$$\mathbb{E}_{h \sim \rho_{\hat{\boldsymbol{w}}}}[L_n(h)] = \frac{1}{n} \sum_{i=1}^{n} \mathbb{E}_{\boldsymbol{w} \sim \rho_{\hat{\boldsymbol{w}}}}[|Y_i - \langle \boldsymbol{w}, X_i \rangle|] = \frac{1}{n} \sum_{i=1}^{n} |Y_i - \langle \hat{\boldsymbol{w}}, X_i \rangle|] = L_n(\hat{h}).$$

Finally let us derive (23). Using Lemma 2 we write

$$\mathbb{E}_{h \sim \rho_{\hat{\boldsymbol{w}}}}[\mathbb{V}_n(h)] = \frac{1}{n-1} \sum_{i=1}^{n} \mathbb{E}_{\boldsymbol{w} \sim \rho_{\hat{\boldsymbol{w}}}}\left[(Y_i - \langle \boldsymbol{w}, X_i \rangle)^2\right] - \frac{1}{n(n-1)} \mathbb{E}_{\boldsymbol{w} \sim \rho_{\hat{\boldsymbol{w}}}}\left[\left(\sum_{i=1}^{n} |Y_i - \langle \boldsymbol{w}, X_i \rangle|\right)^2\right].$$
(24)

Now note that

$$\mathbb{E}_{\boldsymbol{w} \sim \rho_{\hat{\boldsymbol{w}}}}\left[(Y_i - \langle \boldsymbol{w}, X_i \rangle)^2\right] = \mathbb{E}_{\boldsymbol{w} \sim \rho_{\hat{\boldsymbol{w}}}}\left[(Y_i - \langle \hat{\boldsymbol{w}}, X_i \rangle + \langle \hat{\boldsymbol{w}} - \boldsymbol{w}, X_i \rangle)^2\right] =$$
$$= (Y_i - \langle \hat{\boldsymbol{w}}, X_i \rangle)^2 + 2\mathbb{E}_{\boldsymbol{w} \sim \rho_{\hat{\boldsymbol{w}}}}\left[(Y_i - \langle \hat{\boldsymbol{w}}, X_i \rangle)(\langle \hat{\boldsymbol{w}} - \boldsymbol{w}, X_i \rangle)\right] + \mathbb{E}_{\boldsymbol{w} \sim \rho_{\hat{\boldsymbol{w}}}}\left[(\langle \hat{\boldsymbol{w}} - \boldsymbol{w}, X_i \rangle)^2\right].$$

The second summand in the last expression is equal to zero since $\mathbb{E}_{\boldsymbol{w} \sim \rho_{\hat{\boldsymbol{w}}}}[\boldsymbol{w}] = \hat{\boldsymbol{w}}$. By the same reason we conclude that

$$\mathbb{E}_{\boldsymbol{w} \sim \rho_{\hat{\boldsymbol{w}}}}\left[(\langle \hat{\boldsymbol{w}} - \boldsymbol{w}, X_i \rangle)^2\right] = \mathbb{V}_{\boldsymbol{w} \sim \rho_{\hat{\boldsymbol{w}}}}[\langle \hat{\boldsymbol{w}} - \boldsymbol{w}, X_i \rangle].$$

Now note that vector $(\hat{\boldsymbol{w}} - \boldsymbol{w}) \in \mathbb{R}^d$ is uniformly distributed in the $d$-dimensional ball with radius $\hat{\epsilon}$ centred at the origin and also that $\|X_i\|_2 \leq 1$. We can apply Lemma 3 to get

$$\mathbb{E}_{\boldsymbol{w} \sim \rho_{\hat{\boldsymbol{w}}}}\left[(\langle \hat{\boldsymbol{w}} - \boldsymbol{w}, X_i \rangle)^2\right] = \frac{(\hat{\epsilon}\|X_i\|_2)^2}{d+2},$$

meaning that

$$\mathbb{E}_{\boldsymbol{w} \sim \rho_{\hat{\boldsymbol{w}}}}\left[(Y_i - \langle \boldsymbol{w}, X_i \rangle)^2\right] = (Y_i - \langle \hat{\boldsymbol{w}}, X_i \rangle)^2 + \frac{(\hat{\epsilon}\|X_i\|_2)^2}{d+2}.$$
(25)

Finally

$$\mathbb{E}_{\boldsymbol{w} \sim \rho_{\hat{\boldsymbol{w}}}}\left[\left(\sum_{i=1}^{n} |Y_i - \langle \boldsymbol{w}, X_i \rangle|\right)^2\right] = \mathbb{V}_{\boldsymbol{w} \sim \rho_{\hat{\boldsymbol{w}}}}\left[\sum_{i=1}^{n} |Y_i - \langle \boldsymbol{w}, X_i \rangle|\right] + \left(\sum_{i=1}^{n} \mathbb{E}_{\boldsymbol{w} \sim \rho_{\hat{\boldsymbol{w}}}}[|Y_i - \langle \boldsymbol{w}, X_i \rangle|]\right)^2 =$$
$$= \mathbb{V}_{\boldsymbol{w} \sim \rho_{\hat{\boldsymbol{w}}}}\left[\sum_{i=1}^{n} |Y_i - \langle \boldsymbol{w}, X_i \rangle|\right] + \left(n L_n(\hat{h})\right)^2,$$
(26)

where we used (22). For any sequence of random variables $\xi_1, \dots, \xi_n$ we have

$$\mathbb{V}\left[\sum_{i=1}^{n} \xi_i\right] = \sum_{i=1}^{n} \sum_{j=1}^{n} \mathbb{E}\left[(\xi_i - \mathbb{E}[\xi_i])(\xi_j - \mathbb{E}[\xi_j])\right].$$

Using

$$\xi_i = |Y_i - \langle \boldsymbol{w}, X_i \rangle| = |Y_i - \langle \hat{\boldsymbol{w}}, X_i \rangle + \langle \hat{\boldsymbol{w}} - \boldsymbol{w}, X_i \rangle|,$$

we can rewrite

$$\mathbb{V}_{\boldsymbol{w} \sim \rho_{\hat{\boldsymbol{w}}}}\left[\sum_{i=1}^{n} |Y_i - \langle \boldsymbol{w}, X_i \rangle|\right] =$$
$$= \sum_{i=1}^{n} \sum_{j=1}^{n} \mathbb{E}_{\boldsymbol{w} \sim \rho_{\hat{\boldsymbol{w}}}}\left[\left(\xi_i - |Y_i - \langle \hat{\boldsymbol{w}}, X_i \rangle|\right)\left(\xi_j - |Y_j - \langle \hat{\boldsymbol{w}}, X_j \rangle|\right)\right],$$
(27)

where we again used the fact that random variables $Y_i - \langle \boldsymbol{w}, X_i \rangle$, $i = 1, \dots, n$, does not change the sign. Since for any $a, b \in \mathbb{R}$ such that $|b| \geq |a|$ we have $|b + a| - |b| = \mathrm{sgn}\{b\} \cdot a$ we can write

$$\xi_i - |Y_i - \langle \hat{\boldsymbol{w}}, X_i \rangle| = \mathrm{sgn}\{Y_i - \langle \hat{\boldsymbol{w}}, X_i \rangle\}\langle \hat{\boldsymbol{w}} - \boldsymbol{w}, X_i \rangle.$$

Then we have

$$\mathbb{E}_{\boldsymbol{w} \sim \rho_{\hat{w}}}\left[\left(\xi_i - |Y_i - \langle \hat{\boldsymbol{w}}, X_i \rangle|\right)\left(\xi_j - |Y_j - \langle \hat{\boldsymbol{w}}, X_j \rangle|\right)\right] =$$

$$= \operatorname{sgn}\left\{(Y_i - \langle \hat{\boldsymbol{w}}, X_i \rangle)(Y_j - \langle \hat{\boldsymbol{w}}, X_j \rangle)\right\} \mathbb{E}_{\boldsymbol{w} \sim \rho_{\hat{w}}}\left[\langle \hat{\boldsymbol{w}} - \boldsymbol{w}, X_i \rangle \langle \hat{\boldsymbol{w}} - \boldsymbol{w}, X_j \rangle\right]. \qquad (28)$$

Now use the fact that

$$\mathbb{E}_{\boldsymbol{w} \sim \rho_{\hat{w}}}\left[\langle \hat{\boldsymbol{w}} - \boldsymbol{w}, X_i \rangle \langle \hat{\boldsymbol{w}} - \boldsymbol{w}, X_j \rangle\right] =$$

$$= \frac{1}{4}\mathbb{E}_{\boldsymbol{w} \sim \rho_{\hat{w}}}\left[\left(\langle \hat{\boldsymbol{w}} - \boldsymbol{w}, X_i + X_j \rangle\right)^2 - \left(\langle \hat{\boldsymbol{w}} - \boldsymbol{w}, X_i - X_j \rangle\right)^2\right] =$$

$$= \frac{1}{4}\mathbb{V}_{\boldsymbol{w} \sim \rho_{\hat{w}}}\left[\langle \hat{\boldsymbol{w}} - \boldsymbol{w}, X_i + X_j \rangle\right] - \frac{1}{4}\mathbb{V}_{\boldsymbol{w} \sim \rho_{\hat{w}}}\left[\langle \hat{\boldsymbol{w}} - \boldsymbol{w}, X_i - X_j \rangle\right].$$

Again noting that vector $(\hat{\boldsymbol{w}} - \boldsymbol{w}) \in \mathbb{R}^d$ is uniformly distributed in the $d$-dimensional ball with radius $\hat{\epsilon}$ centred at the origin and that $\|X_i - X_j\|_2 \leq 1$, $\|X_i + X_j\|_2 \leq 1$, we can apply Lemma 3 and get

$$\mathbb{E}_{\boldsymbol{w} \sim \rho_{\hat{w}}}\left[\langle \hat{\boldsymbol{w}} - \boldsymbol{w}, X_i \rangle \langle \hat{\boldsymbol{w}} - \boldsymbol{w}, X_j \rangle\right] =$$

$$= \frac{(\hat{\epsilon}\|X_i + X_j\|_2)^2}{4(d+2)} - \frac{(\hat{\epsilon}\|X_i - X_j\|_2)^2}{4(d+2)} = \frac{\hat{\epsilon}^2 \langle X_i, X_j \rangle}{4(d+2)}. \qquad (29)$$

Combining (24)–(29) altogether we complete the proof. $\qquad \square$

Note that the choice $\hat{\epsilon} = \min_{i=1,\ldots,n}|Y_i - \langle \hat{\boldsymbol{w}}, X_i \rangle|$ can lead to very large values of $\mathrm{KL}(\rho_{\hat{w}}, \pi)$ because of the equation (21). We can overcome this problem using the following theorem which lets us pick arbitrary value of $\epsilon$.

**Theorem 7.** *Let $n_\epsilon$ be the number of points such that $|Y_i - \langle \hat{\boldsymbol{w}}, X_i \rangle| < \epsilon$. Then for posterior and prior distributions $\rho_{\hat{w}}$ and $\pi$ defined as in Section 5 we have*

$$\mathrm{KL}(\rho_{\hat{w}}\|\pi) = d\ln\frac{2}{\epsilon};$$

$$\mathbb{E}_{h \sim \rho_{\hat{w}}}[L_n(h)] \leq L_n(h_{\hat{w}}) + \epsilon\frac{n_\epsilon}{n};$$

$$\mathbb{E}_{h \sim \rho_{\hat{w}}}[\mathbb{V}_n(h)] \leq \frac{1}{n-1}\sum_{i=1}^{n}\left((Y_i - \langle \hat{\boldsymbol{w}}, X_i \rangle)^2 + \frac{\epsilon^2}{d+2}\|X_i\|_2^2\right) -$$

$$- \frac{1}{n(n-1)}\left(\sum_{i=1}^{n}\mathbb{1}\{|Y_i - \langle \hat{\boldsymbol{w}}, X_i \rangle| \geq \epsilon\}\left[|Y_i - \langle \hat{\boldsymbol{w}}, X_i \rangle|\right]\right)^2 - \frac{1}{n(n-1)}\sum_{i=1}^{n}\sum_{j=1}^{n}\gamma_{i,j};$$

$$\gamma_{i,j} = \begin{cases} \operatorname{sgn}\left\{(Y_i - \langle \hat{\boldsymbol{w}}, X_i \rangle)(Y_j - \langle \hat{\boldsymbol{w}}, X_j \rangle)\right\}\frac{\epsilon^2\langle X_i, X_j \rangle}{4(d+2)}, & \text{if } A_i \cap A_j; \\ -\frac{\epsilon^2\|X_i\|_2}{\sqrt{d+2}}, & \text{if } A_i \cap A_j^c; \\ -\frac{\epsilon^2\|X_j\|_2}{\sqrt{d+2}}, & \text{if } A_i^c \cap A_j; \\ -\epsilon^2, & \text{if } A_i^c \cap A_j^c. \end{cases}$$

*where we have defined events $A_i = \{|Y_i - \langle \hat{\boldsymbol{w}}, X_i \rangle| \geq \epsilon\}$ and $A^c$ is the complement of event $A$.*

*Proof.* The proof repeats the one of Theorem 6 with minor changes. The main difference is that now for indices $i$ such that $|Y_i - \langle \hat{\boldsymbol{w}}, X_i \rangle| < \epsilon$ random variables $\xi_i = Y_i - \langle \boldsymbol{w}, X_i \rangle$ change their signs as $\boldsymbol{w}$ varies. Thus for these $\xi_i$ the mean value $\mathbb{E}[|\xi_i|]$ is no longer $|Y_i - \langle \hat{\boldsymbol{w}}, X_i \rangle|$ and has more complicated form. Instead of computing $\mathbb{E}_{h \sim \rho_{\hat{w}}}[L_n(h)]$ precisely we will upper bound $\mathbb{E}[|\xi_i|]$ for such a $i$ using

$$\mathbb{E}_{\boldsymbol{w} \sim \rho_{\hat{w}}}[|Y_i - \langle \boldsymbol{w}, X_i \rangle|] = \mathbb{E}_{\boldsymbol{w} \sim \rho_{\hat{w}}}[|Y_i - \langle \hat{\boldsymbol{w}}, X_i \rangle + \langle \hat{\boldsymbol{w}} - \boldsymbol{w}, X_i \rangle|] \leq$$

$$\leq |Y_i - \langle \hat{\boldsymbol{w}}, X_i \rangle| + \epsilon\|X_i\|_2 \leq |Y_i - \langle \hat{\boldsymbol{w}}, X_i \rangle| + \epsilon$$

which completes the proof for the $\mathbb{E}_{h \sim \rho_{\hat{w}}}[L_n(h)]$.

We will also derive an upper bound for $\mathbb{E}_{h \sim \rho_{\hat{w}}}[\mathbb{V}_n(h)]$. To do so we need the lower bound for the second term in right hand side of (24) (first term stays unchanged compared to Theorem 6). First we will use the following lower bound for the term appearing in (26):

$$\left( \sum_{i=1}^{n} \mathbb{E}_{\boldsymbol{w} \sim \rho_{\hat{w}}}[|Y_i - \langle \boldsymbol{w}, X_i \rangle|] \right)^2 \geq \left( \sum_{i=1}^{n} \mathbb{1}\{|Y_i - \langle \hat{\boldsymbol{w}}, X_i \rangle| \geq \epsilon\}[|Y_i - \langle \hat{\boldsymbol{w}}, X_i \rangle|] \right)^2.$$

Finally we need the lower bound for the variance

$$\mathbb{V}_{\boldsymbol{w} \sim \rho_{\hat{w}}} \left[ \sum_{i=1}^{n} |Y_i - \langle \boldsymbol{w}, X_i \rangle| \right]$$

which we derive through the lower bounds for the covariance terms appearing in (27) corresponding to the pairs $(i, j)$ such that either $|Y_i - \langle \hat{\boldsymbol{w}}, X_i \rangle| < \epsilon$ or $|Y_j - \langle \hat{\boldsymbol{w}}, X_j \rangle| < \epsilon$ (for all the other terms we have already derived the precise forms in previous proof). It is known that for random variables $\xi, \eta$ of finite variances the following holds:

$$\left| \mathbb{E}\big[(\xi - \mathbb{E}[\xi])(\eta - \mathbb{E}[\eta])\big] \right| \leq \sqrt{\mathbb{V}[\xi]\mathbb{V}[\eta]},$$

meaning

$$\mathbb{E}\big[(\xi - \mathbb{E}[\xi])(\eta - \mathbb{E}[\eta])\big] \geq -\sqrt{\mathbb{V}[\xi]\mathbb{V}[\eta]}.$$

Note that if $|Y_i - \langle \hat{\boldsymbol{w}}, X_i \rangle| < \epsilon$ then $|Y_i - \langle \boldsymbol{w}, X_i \rangle| \leq 2\epsilon$ and we have

$$\mathbb{V}_{\boldsymbol{w} \sim \rho_{\hat{w}}}[|Y_i - \langle \boldsymbol{w}, X_i \rangle|] \leq \epsilon^2,$$

where we used the fact that for random variable $\xi \in [0, 1]$ we have $\mathbb{V}[\xi] \leq 1/4$.

If $|Y_i - \langle \hat{\boldsymbol{w}}, X_i \rangle| \geq \epsilon$ we have

$$\mathbb{V}_{\boldsymbol{w} \sim \rho_{\hat{w}}}[|Y_i - \langle \boldsymbol{w}, X_i \rangle|] = \mathbb{V}_{\boldsymbol{w} \sim \rho_{\hat{w}}}[\langle \hat{\boldsymbol{w}} - \boldsymbol{w}, X_i \rangle] = \frac{(\epsilon \|X_i\|_2)^2}{d+2}$$

which completes the proof. $\qquad\square$

**Comments on section 5.** Note that for posterior distribution $\rho_{\hat{w}}$ defined as in Section 5 we clearly have $B(x, \rho_{\hat{w}}) = \mathbb{E}_{\boldsymbol{w} \sim \rho_{\hat{w}}}[h_{\boldsymbol{w}}(x)] = h_{\hat{\boldsymbol{w}}}(x)$ meaning that the weighted (Bayes) regression rule coincides with the deterministic hypothesis $h_{\hat{\boldsymbol{w}}}$. Also note that the convexity of absolute deviation loss implies that

$$\mathbb{E}_{(X,Y) \sim \mathcal{D}}\big[|B(X, \rho) - Y|\big] \leq \mathbb{E}_{(X,Y) \sim \mathcal{D}}\mathbb{E}_{\boldsymbol{w} \sim \rho}\big[|h_{\boldsymbol{w}}(X) - Y|\big] = L(G_\rho)$$

for any distribution $\rho$. Together these two facts yield that any upper bound on the true loss of Gibbs regression rule associated with posterior distribution $\rho_{\hat{w}}$ also upper bounds the true loss of deterministic hypothesis $h_{\hat{\boldsymbol{w}}}$. The same is true if we use quadratic or any other convex loss function instead of the absolute loss.

## C  PAC-Bayesian Lemma

*Proof of Lemma 1.* We start with Donsker-Varadhan's variational definition of relative entropy [17], which states that $\mathrm{KL}(\rho\|\pi) = \sup_f \left( \mathbb{E}_\rho[f(h)] + \ln \mathbb{E}_\pi[e^{f(h)}] \right)$, where the supremum is taken over all measurable functions $f : \mathcal{H} \to \mathbb{R}$. Obviously, we can extend the range of $f$ and take $f = f_n : \mathcal{H} \times (\mathcal{X} \times \mathcal{Y})^n \to \mathbb{R}$. Changing sides in the definition we have that

$$\mathbb{E}_\rho[f_n(h, S)] \leq \mathrm{KL}(\rho\|\pi) + \ln \mathbb{E}_\pi\left[e^{f_n(h,S)}\right] \tag{30}$$

for all pairs $(\rho, \pi)$ simultaneously and any $S$. Note that there is nothing probabilistic in the above argument.

Now we fix $\pi$ (so that $\pi$ does not depend on $S$). Then with probability greater than $1 - \delta$ (over the randomness of the data) we have:

$$\mathbb{E}_\pi\left[e^{f_n(h,S)}\right] \leq \frac{1}{\delta}\mathbb{E}_{S' \sim \mathcal{D}^n}\left[\mathbb{E}_{h \sim \pi}\left[e^{f_n(h,S')}\right]\right] = \frac{1}{\delta}\mathbb{E}_{h \sim \pi}\left[\mathbb{E}_{S' \sim \mathcal{D}^n}\left[e^{f_n(h,S')}\right]\right],$$

where the first step follows by Markov's inequality (we consider $\mathbb{E}_\pi\left[e^{f_n(h,S)}\right]$ as a random variable and apply Markov's inequality to this random variable) and in the second step we can exchange the order of expectations because $\pi$ is independent of $S$. Substituting this result back into (30) we obtain that with probability greater than $1 - \delta$ simultaneously for all $\rho$:

$$\mathbb{E}_\rho\left[f_n(h,S)\right] \leq \mathrm{KL}(\rho\|\pi) + \ln\frac{1}{\delta} + \ln\mathbb{E}_\pi\left[\mathbb{E}_{S'\sim\mathcal{D}^n}\left[e^{f_n(h,S')}\right]\right].$$

$\square$