[Reviews · NeurIPS 2013]

Submitted by Assigned_Reviewer_4

The paper present uniform convergence bounds which combines the PAC-Bayessian technique with empirical Bernstein inequalities. This allows the authors to improve upon the state-of-the-art under some regimes of parameters.

The paper takes a well studied problem (uniform convergence) and improves on top of the best known results. It is well presented and I like the fact that the authors clearly explain the regimes in which the new result is better than the current state-of-the-art and the regimes in which it is slightly worse. The presentation is very clear and easy to follow.

One place that I think the authors can improve is in exact definition of the empirical variance $E_\rho[V_n(h)]$. While $V_n(h)$ is defined in (5), how to take the expectation is not very clear.
Summary: The authors address a classical problem in machine learning: uniform convergence, and improve upon the best known results.

Submitted by Assigned_Reviewer_5

PAC-Bayes-Empirical Bernstein Inequality

The paper proposes a new PAC-Bayesian bound which has two specific features: a) it
uses empirical estimate of the variance of the Gibbs classiier and b) it combines
PAC-Bayesian techniques with concentration inequalities derived for self-bounding functions.
The new result is a "refinement" of a result by Seldin et al. on PAC-Bayesian bound
using the actual variance. Empirical results support the relevance of the approach.

The paper tackles an important question: that of having at hand accurate (PAC-Bayesian) bound
based on quantities that can be estimated from data. Here the authors propose a nice
approach to derive such a bound in the PAC-Bayesian framework. The technical results are
correct, the write-up is good and the empirical results show that there are situations where
the provided bound is indeed more accurate than other existing bounds.

It essentially a good and solid paper.

I have two questions:
- what would be the difficulty of going one step further and provide PAC-Bayesian bounds for, e.g. U-statistics, which are very important in the case of e.g. ranking, and for which there exist one the one hand empirical Bernstein inequalities and, on the other hand, dedicated PAC-Bayesian bounds ?
- more generally, what about non-IID settings ?
Summary: Good paper introducing an original generalization bound that makes use of quantities computable from data. Nice writing and empirical evaluation.

Submitted by Assigned_Reviewer_7

This paper derives a new empircal PAC Bayesian bound by combining an existing (non-empircal) PAC Bayesian Berstein bound (i.e., involving the true variance of the loss values) with a PAC Bayesian analysis of the concentration of the empirical variance around its true value. This new bound has the advantage of being tighter when the empirical variance is small compared to the empirical loss. Experiments on real and empirical data with simple models compare the new bound with the usual empirical PAC Bayesian bound confirming the advantage.

While this is a well written paper that make a novel contribution to the PAC Bayesian literature it is a incremental and only marginally significant one. As the authors' claim, the newer bound is only really useful for analysing cases when there is a large mismatch between the model class and the data generating distribution.

Although the idea of applying a PAC Bayesian bound to the difference between empirical and true variances is original and addresses a shortcoming in the existing work, it appears that the steps from there were fairly routine, leaning heavily on techniques developed in other recent papers. That said, the definitions and proofs all seem to be correct and are logically organised.

Minor suggestions:

1. Put an article ("a" or "the") before all singular uses of "PAC-Bayes-Empirical-Berstein inequality" and the like. For example, the opening sentence of the abstract should be "We present a PAC-Bayes-Empirical-Bernstein inequality..."

2. You should define $KL(\rho\|\pi)$ or otherwise make clear in which direction the relative entropy is defined.
Summary: A well motivated, clearly written, novel but modest contribution to the PAC Bayesian literature. I was not particularly surprised by any of the results or implications in this paper.
Author Feedback

Author rebuttal: We would like to thank the reviewers for their positive reviews, useful comments, and insightful questions.

Below we answer the questions raised by the reviewers. Reviewers are addressed by their number and their comments begin with “>”.

To Rev. 4
------------
> One place that I think the authors can improve is in exact definition of the empirical variance $E_\rho[V_n(h)]$. While $V_n(h)$ is defined in (5), how to take the expectation is not very clear.

The exact definition of $E_\rho[V_n(h)]$ is $E_{h\sim \rho}[V_n(h)]$, analogously to $E_\rho [L_n(h)]$. We thank the reviewer for pointing out that we forgot to put this definition and will add it to the revised manuscript.

To Rev. 5
------------
> what would be the difficulty of going one step further and provide PAC-­Bayesian bounds for, e.g. U­ statistics, which are very important in the case of e.g. ranking, and for which there exist on the one hand empirical Bernstein inequalities and, on the other hand, dedicated PAC­ Bayesian bounds?

We note that variance is second-order U-statistics and our bound in Theorem 3 treats this case. For some other U-statistics we would first of all need a bound on its moment generating function. This is definitely not a trivial task, but for many U-statistics we already have it (as it was in our case for the variance). Then it is possible to follow the general framework, as we did, up to equation (9). The main difficulty is the final step after the substitution of the bound on moment generating function in equation (9). At the moment we have no general recipe on how to finish the calculation after the substitution and we suspect that each case may require individual treatment. In our case it took 3 pages of calculations and we assume that for higher-order U-statistics it may be even more involved.

> more generally, what about non­ IID settings?

At the moment we are aware of two ways of extending IID to non-IID settings. One way is extension to martingales, as done by Seldin et. al. (2012). Our results can be directly extended to martingales in a similar way. The second way is picking an IID subsample from a dependent sample and working with the IID subsample (see Ralaivola et. al. (2010)). In this case the sample size n in the bound reduces to the size of IID subsample, but otherwise the results are identical.

To Rev. 7
------------
> Although the idea of applying a PAC Bayesian bound to the difference between empirical and true variances is original and addresses a shortcoming in the existing work, it appears that the steps from there were fairly routine, leaning heavily on techniques developed in other recent papers.

We are happy that things looked simple to the reviewer, but we would like to note that although we followed a general framework, putting all pieces together was not a “fairly routine” procedure and it actually took us 6 pages of non-trivial calculations that we moved to the supplementary material to ease on the reader.

We thank the reviewer for the minor language and technical suggestions that will be incorporated in the revised manuscript.